# Effect of Low Protein Diet Supplemented with Ketoanalogs on Endothelial Function and Protein-Bound Uremic Toxins in Patients with Chronic Kidney Disease

**DOI:** 10.3390/biomedicines11051312

**Published:** 2023-04-28

**Authors:** George Chang, Hong-Mou Shih, Chi-Feng Pan, Chih-Jen Wu, Cheng-Jui Lin

**Affiliations:** 1Division of Nephrology, Department of Internal Medicine, MacKay Memorial Hospital, Taipei 104217, Taiwan; 2Graduate Institute of Physiology, College of Medicine, National Taiwan University, Taipei 100001, Taiwan; 3Department of Medicine, Mackay Medical College, New Taipei 220001, Taiwan; 4Department of Medicine, Mackay Junior College of Medicine, Nursing and Management, Taipei 100001, Taiwan

**Keywords:** chronic kidney disease, ketoanalogs, p-cresyl sulfate, indoxyl sulfate, flow-mediated dilation

## Abstract

Studies have demonstrated that a low-protein diet supplemented with ketoanalogs (KAs) could significantly retard progression of renal function in patients with chronic kidney disease (CKD) stages 3–5. However, its effects on endothelial function and serum levels of protein-bound uremic toxins remain elusive. Therefore, this study explored whether a low-protein diet (LPD) supplemented with KAs affects kidney function, endothelial function, and serum uremic toxin levels in a CKD-based cohort. In this retrospective cohort, we enrolled 22 stable CKD stage 3b–4 patients on LPD (0.6–0.8 g/day). Patients were categorized into control (LPD only) and study groups (LPD + KAs 6 tab/day). Serum biochemistry, total/free indoxyl sulfate (TIS/FIS), total/free p-cresyl sulfate (TPCS/FPCS), and flow-mediated dilation (FMD) were measured before and after 6 months of KA supplementation. Before the trial, there were no significant differences in kidney function, FMD, or uremic toxin levels between the control and study groups. When compared with the control group, the paired *t*-test showed a significant decrease in TIS and FIS (all *p* < 0.05) and a significant increase in FMD, eGFR, and bicarbonate (all *p* < 0.05). In multivariate regression analysis, an increase in FMD (*p* < 0.001) and a decrease in FPCS (*p* = 0.012) and TIS (*p* < 0.001) remained persistent findings when adjusted for age, systolic blood pressure (SBP), sodium, albumin, and diastolic blood pressure (DBP). LPD supplemented with KAs significantly preserves kidney function and provides additional benefits on endothelial function and protein-bound uremic toxins in patients with CKD.

## 1. Introduction

Globally, rapidly, and with increasing prevalence, chronic kidney disease (CKD) is becoming one of the leading public health problems, affecting 13.4% of the adult population [1], and is also a leading cause of death in the global burden of disease studies [2].

CKD patients are at a higher risk for developing cardiovascular disease (CVD), which, in turn, is associated with a more rapid progression of CKD [3]. In comparison to the general population, patients with CKD are more susceptible to developing CVD because they are more likely to have both traditional risk factors (e.g., hypertension, diabetes, advanced age, dyslipidemia, and tobacco use) and particularly nontraditional risk factors (e.g., anemia, proteinuria, pro-inflammatory cytokines, oxidative stress, hyperhomocysteinemia, abnormal calcium and phosphate metabolism, and accumulation of uremic toxin) that are linked to CVD [4]. Studies have shown an inverse association between estimated glomerular filtration rate (eGFR) and cardiovascular risk [5], and a meta-analysis of 19 creatinine-based studies has shown that for each 30% reduction in eGFR, the risk of major vascular events increases by about 30% [6].

CKD is associated with significant morbidity and mortality; hence the management of this condition is crucial. This involves prioritizing key aspects such as mitigating decline in glomerular filtration rate, preventing cardiovascular diseases, and managing associated complications. Currently, there are a few strategies and treatments accessible for deterring CKD progression and minimizing cardiovascular incidents, including lifestyle adjustments, regulating blood pressure and blood sugar levels, employing RAS inhibitors, and more recently, incorporating SGLT2 inhibitors [7]

Despite optimal management, the risk of CKD progression remains [8], and novel management options are still being explored. A restricted protein diet has long been shown to preserve residual renal mass and slow CKD progression, with the pathophysiology of mitigating glomerular hyperfiltration and reducing uremic toxins [9]. Adding ketoanalogs (KAs) to a very low-protein diet (sVLPD) was essential to fulfill minimal protein requirement and prevent cachexia. It has been shown in several clinical trials to retard the progression of CKD [10] and improve metabolic markers [11,12]. More recently, trials have shown the benefits of KAs in addition to a low-protein diet (LPD), with delayed initiation of maintenance dialysis [13,14]. Various systemic reviews and meta-analyses have also emerged, which support the benefit of supplementing KAs to an LPD in patients with moderate to advanced CKD [15,16], even in patients with diabetic kidney disease [17], with results demonstrating delayed CKD progression without causing malnutrition. As renal function deteriorates, uremic toxins, which are actively secreted via the renal tubule, gradually accumulate. It negatively affects almost every organ, most notably the cardiovascular system [18,19] and represents one area of nontraditional cardiovascular risk factors unique to patients with CKD. Recently, protein-bound uremic toxins, indoxyl sulfate (IS), and p-cresol sulfate (PCS) have been among the most extensively studied.

Studies have shown that PCS and IS have not only a positive association between each other in CKD patients, but also a correlation with CKD stages. As eGFR decreases, renal clearance of IS and PCS decreases, and total plasma levels rise in proportion to the eGFR [20,21], reaching their peak at the stage of hemodialysis. In addition, IS and PCS are valuable markers for predicting cardiovascular events and renal function progression in CKD patients [22,23]. More importantly, elevated levels of PCS and IS are associated with increased mortality in patients with CKD, while PCS, but not IS, is associated with an increased risk of cardiovascular events [24]. One of the most recent theories regarding the effect of KA supplementation is the ability to simultaneously reduce protein intake and modulate intestinal microbiota, resulting in both IS and PCS reduction [25,26].

As previously mentioned, IS and PCS have been associated with the progression of CKD. Putative mechanisms include inflammatory and fibrotic pathways which induce renal toxicity [27], with endothelial cells being their predominant target. Previous in vitro studies have also shown the concentration-dependent inhibitory effect of IS on both motility and colony formation of endothelial progenitor cells, as well as its strong correlation with vascular dysfunction, providing at least one plausible explanation for its clinical relevance [28].

Furthermore, and as mentioned earlier, patients diagnosed with chronic kidney disease (CKD) are more susceptible to both traditional and nontraditional cardiovascular risk factors [29]. The risk factors for cardiovascular disease are also frequently associated with the risk factors for endothelial dysfunction. Additionally, certain medications such as antihypertensives [30] and erythropoiesis-stimulating agents (ESA) [31] can also have an impact on the endothelium. Prolonged exposure to cardiovascular risk factors and oxidative stress overwhelms the defensive mechanisms of the vascular endothelium, resulting in endothelial dysfunction, loss of endothelial integrity, increased proliferation and migration of smooth muscle cells, and enhanced adhesion and migration of leukocytes [29]. Reduced nitric oxide (NO) bioavailability [32] is a hallmark of endothelial dysfunction in CKD, and clinically, noninvasive assessment of endothelial function is commonly performed using flow-mediated dilation (FMD) technique.

The majority of studies have demonstrated an inverse relationship between FMD and the likelihood of future cardiovascular events [33]. In particular, a significant negative correlation was noted between brachial FMD and cardiovascular risk in Asian populations [34]. Currently, there are limited studies investigating the effect of KAs with LPD on the level of IS, PCS, and endothelial functions.

This study investigated whether adding KAs to CKD patients on an LPD could reduce the level of uremic toxins and result in reduced endothelial dysfunction (measured via FMD), thereby abating the progression of CKD.

## 2. Materials and Methods

### 2.1. Study Design and Subjects

This was a single-center, retrospective cohort study, conducted for 6 months. Twenty-two stable patients with CKD stages 3b and 4, who had been on an LPD for the past 6 months, were enrolled from the outpatient department at Mackay Memorial Hospital. All of the patients were taking either ACE inhibitors (ACEI) or angiotensin receptor blockers (ARB) as their antihypertensive medications. Due to the restricted size of our cohort, individuals who were smokers or were with diabetes and iron deficiency anemia (IDA) were excluded to minimize the extent of confounding, particularly with regard to endothelial functions. Patients were classified into two groups at a ratio of 1:1, with and without using KAs. A control group of 11 patients was compared with the study group of 11 patients who took Ketosteril 6 tabs/day. Data were obtained before and after 6 months for both groups. The primary outcome was the level of uremic toxins (IS and PS), while the secondary outcome was FMD measurements. The study was conducted in accordance with the Declaration of Helsinki and approved by the ethics committee of the Mackay Memorial Hospital. All participants signed an informed consent approved by the Institutional Review Board (protocol code 20MMHIS426e, 26 May 2022) for studies involving humans.

### 2.2. FMD Measurements

FMD of the brachial artery was performed using an ALOKA ultrasound machine (Hitachi Aloka Medical, Ltd., Tokyo, Japan) according to the guidelines set by the International Brachial Artery Reactivity Task Force. The brachial artery was longitudinally imaged above the elbow using an 11.3-MHz probe. After the image was recorded for 2 min, forearm ischemia was induced by the rapid deflation of a cuff that was inflated to 200 mmHg (or 50 mmHg above SBP, whichever was higher) below the elbow for 5 min. The resulting reactive hyperemia was recorded for an additional 2 min. A single trained investigator used Brachial Analyzer software (v.5.0, Medical Imaging Applications, Coralville, IA, USA) to analyze all FMDs.

### 2.3. Statistical Analysis

Demographic data are expressed as mean ± SD, which was used to present the similarities between the study and control groups, and the value > 0.05 indicated insignificant differences in covariates between them. Paired *t*-tests were performed between pre-test and post-test (after 6 months) for the control and study groups, where variables with a significant difference were identified. A *t*-test was then performed between the study and control groups to determine the variables with a significant difference after 6 months. Multivariate regression analysis was performed to determine whether the result persisted after adjusting for baseline covariates to determine the effects of KA (Ketosteril) on FMD and uremic toxins. Statistical significance was defined as a two-sided *p*-value of < 0.05. All statistical analyses were conducted using the SPSS software programs (v.17.0, SPSS Inc., Chicago, IL, USA).

## 3. Results

Twenty-two stable patients with CKD stages 3b to 4 who had been on an LPD for the past 6 months were enrolled in the study. Before the study, we did not observe a significant difference in the baseline characteristics of renal function, uremic toxin levels, lipid profile, electrolytes, and bicarbonate levels. However, significant differences in age (*p* = 0.037), blood pressure (*p* < 0.001), Na (*p* = 0.02), hemoglobin (*p* = 0.046), and albumin (*p* = 0.044) were observed (Table 1). No significant differences were observed in the hematocrit (Hct) level. The control group had more elderly patients with higher blood pressure, lower albumin, hemoglobin and sodium levels than the study group.

Paired *t*-tests were performed between the pre-test and post-test for the study and control groups, with a final comparison between the two to determine whether the mean difference between the two sets of observations was zero (Table 2).

In the study group, increases in FMD (4.54 ± 0.64), eGFR (25.42 ± 11.28), and HCO3 (22.85 ± 1.59), with decreases in total PCS (9.73 ± 11.03) and total IS (8.15 ± 9.69) were noted with a significant difference between pre-and post-test (Figure 1).

As for the control group, results revealed significant differences between the pre-test and post-test with a decrease in FMD (3.38 ± 0.43), GFR (16.69 ± 3.96), Ca (8.48 ± 0.38), albumin (3.79 ± 0.27), and HCO3 (18.73 ± 3.04), and an increase in FPCS (0.45 ± 0.34), TIS (6.32 ± 1.91), FIS (0.29 ± 0.13), BUN (59.60 ± 12.75), Cr (4.01 ± 1.11) and P (4.73 ± 0.54). (Figure 2).

Finally, when we compared the study and control groups, the *t*-test showed a significant decrease in TIS, FIS, TPCS, FPCS, TG, and P, with a significant increase in FMD, eGFR, and HCO3 (Figure 3).

We adjusted our primary and secondary outcomes for baseline status using a multivariate regression analysis. When we adjusted for age, SBP, Na, albumin, and DBP, increases in FMD (0.91, 0.50–1.41, *p* < 0.001) and decreases in FPCS (−0.37, −0.65–−0.09, *p* < 0.012) and TIS (−3.23, −4.61–−1.84, *p* < 0.001) remained consistent findings (Table 3).

## 4. Discussion

Our findings showed that patients with CKD stages 3b to 4, who were already on LPD and supplemented with KAs showed a significant reduction in IS and PCS with improved FMD compared to those without after 6 months. Confounding was handled by adjusting the baseline covariates, where the result remained persistent. Simultaneously, a decrease in eGFR and improved metabolic profiles, such as bicarbonate, Ca, and P levels were observed. To our knowledge, this is the first study to focus on the effect of KAs on both uremic toxin levels and endothelial function in patients with CKD3b-4 who were already on LPD.

We would like to draw attention to the notable discrepancy in hemoglobin levels at baseline between the study and control groups. The disparity was only marginally significant (*p* = 0.046), and the hematocrit levels were within the normal range. Consequently, we considered the difference to be insignificant and excluded it as a covariate in our multivariate regression analysis.

Anemia is recognized to affect endothelial function and is considered one of the nontraditional risk factors for endothelial dysfunction. Red blood cells (RBCs) play a crucial role in transporting NO, synthesizing NO through nitrite reduction in hypoxic conditions, and generating NO via active endothelial nitric oxide synthase (eNOS) located within the RBC membrane during normoxia. Impaired eNOS has been identified as a significant factor contributing to RBC dysfunction in both anemic mice and patients with acute coronary syndrome [35].

Research investigating the impact of anemia on endothelial function has primarily concentrated on sickle cell anemia and iron deficiency anemia (IDA). Studies have revealed that RBCs in IDA patients exhibit notable characteristics such as reduced size and increased stiffness. Computational simulations have demonstrated that these altered RBCs tend to accumulate near the vessel wall, resulting in aberrant shear forces that can adversely affect endothelial function [36]. Thus, patients with IDA were excluded from our study from the beginning.

Erythropoiesis-stimulating agents (ESAs) have been demonstrated to influence endothelial function. A prospective study has shown that the administration of ESAs resulted in clinically improved endothelial dysfunction in anemic patients with chronic kidney disease (CKD), primarily through alleviating anemia [31]. However, ESAs were not utilized in our study, as the National Health Insurance (NHI) coverage in Taiwan for ESA only extends to CKD stage 5 and beyond.

As stated earlier, our study excluded smokers and individuals with diabetes. Diabetes is known to impact endothelial function, predominantly through prolonged exposure to hyperglycemia. The underlying mechanisms of endothelial dysfunction due to hyperglycemia are diverse and include: (1) increased formation of advanced glycation end products (AGEs) within cells; (2) activation of various protein kinase C (PKC) isoforms; and (3) induction of nuclear factor-κB (NFkB) [37]. These mechanisms ultimately increase oxidative stress, leading to apoptosis and vascular permeability. The resultant imbalance in vascular homeostasis, due to increased vasoconstriction and impaired vasorelaxation, ultimately fosters diabetic endothelial dysfunction [38]. In future studies, it would be interesting to compare the effect of KAs on endothelial functions between diabetic and non-diabetic CKD patients.

Studies on the dietary effect on CKD progression have shown that for patients with an eGFR < 30 who were on a VLPD, the addition of KAs has demonstrated retarding of renal progression and delaying of dialysis [39]. Moreover, research has shown that the benefits persist even when eGFR is <15 [13]. Recent studies have also extended the benefits of KAs’ addition to LPD, including decreased major adverse cardiovascular events [40], and delayed initiation of dialysis [14] among CKD patients. However, little evidence exists regarding its pathogenesis, and we postulate that using KAs may decrease uremic toxins, ameliorate endothelial dysfunction, and thereby slow the progression of CKD.

As mentioned previously, the most widely explored uremic toxins are PCS and IS. Intestinal anaerobes generate IS during the partial breakdown of tyrosine and phenylalanine, while IS results from the breakdown of tryptophan [41]. They are both the end products of metabolism in the colon. They are metabolized to the sulfate form by the liver, followed by secretion via the Organic Anion Transporter (OAT1/3) on the proximal tubular cells [42,43]. As CKD progresses, IS and PCS accumulate, which results in further deterioration of renal function and contributes to the vicious cycle. Thus, it would seem prudent to investigate methods for lowering them in an attempt to arrest the cycle.

When we compared the study and control groups after 6 months, a significant difference was found in IS, PCS, FMD, GFR, Ca, P, and bicarbonate levels. A reduction in IS and PCS was foreseeable in the study group, most likely because of the use of KAs and the associated reduction in protein intake. The MEDIKA study showed that the levels of IS and PCS were reduced [25] with the use of KAs and VLPD compared to a Mediterranean diet or free diet; however, the benefits were attributed more to the LPD than to the use of KAs [26], as concluded in MEDIKA2. IS and PCS are crucial in the pathogenesis of endothelial dysfunction [44], and FMD acts as a surrogate marker. Our results showed that by supplementing LPDs with KAs, a reduction in IS and PCS was observed and, at the same time, amelioration of the degree of endothelial dysfunction was achieved.

We also observed an increase in FMD in the study group compared with that in the control group. Endothelial dysfunction constitutes the first step of atherosclerosis and vascular calcification, it has shown not only contributing to the development of cardiovascular disease [45], but also to CKD progression [46].

Endothelial dysfunction, induced by IS and PCS, are associated with nephrotoxicity. These substances compromise intercellular junctions by inducing the phosphorylation of vascular endothelium cadherin (VE-cadherin), which leads to disassembly and cell internalization [47]. Subsequently, there is an increase in the permeability of blood vessels and leakage [48]. Additionally, they enhanced the communication between leukocytes and endothelial cells, possibly through degradation of the endothelial glycocalyx [49]. As a consequence, the production of cell adhesion molecules such as E-selectin, vascular cell adhesion molecule 1 (VCAM-1), and intercellular adhesion molecule 1 (ICAM-1) increased, leading to escalated inflammation and atherogenesis [50,51]. IS and PCS were also found to increase the number of endothelial microparticles and alter their microRNA content in vitro [52], which can lead to calcification and osteogenesis [53] in vascular smooth muscle. Furthermore, endothelial microparticles were observed to hinder endothelial repair and proliferation in vitro [52]. Indeed, poor microvascular endothelial function is associated with CKD progression as well as albuminuria [54].

In advanced CKD, IS and PCS impairment of endothelial-dependent vasodilatation is well established, with decreased expression of eNOS and reduction in the bioavailability of NO as the dominant pathophysiology [55]. Furthermore, endothelial dysfunction can be quantified by the degree of FMD of the brachial artery, which reflects the amount of NO production [56]. Therefore, we presume that the increase in FMD may be due to the decrease in IS and PCS and, to some degree, contribute to endothelial dysfunction.

The study group had a significantly higher eGFR after 6 months. Indeed, the impaired endothelial function has been associated with albuminuria and CKD progression independently of blood pressure and diabetes [54]. Therefore, we postulate that a higher eGFR was related to reduced endothelial damage, which was possibly related to the reduction in IS and PCS.

Furthermore, IS and PCS manifest their nephrotoxic effects through multiple alternative mechanisms. IS and PCS are recognized for causing degradation of the remaining renal nephrons, particularly in the proximal tubular cells. This leads to the stimulation of glomerular sclerosis, renal fibrosis, and CKD progression. As a result, the expression of pro-a1 collagen, transforming growth factor b1 (TGF-b1), and tissue inhibitor of metalloproteinase 1 (TIMP-1) genes are increased, leading to a greater loss of nephrons. This, in turn, worsens the progression of CKD [57,58]. Additionally, studies have also indicated a rise in the levels of pro-inflammatory cytokines such as IL-6, which is associated with coronary artery diseases and vascular injury, thus contributing to accentuated CKD progression and mortality in this population [59,60]. Furthermore, elevated concentrations of IS and PCS can trigger renal and vascular cell senescence, along with an increase in reactive oxygen species (ROS), leading to an escalation of oxidative stress [61,62].

The improved mineral metabolism (higher calcium and lower serum phosphate) was significantly increased after 6 months in the study group. This is likely due to the calcium content in the KAs which allows sufficient calcium content while mitigating the risks of hyperphosphatemia [63].

Moreover, we also observed an increase in serum bicarbonate, which could be related to both a reduction in protein intake and a diet rich in fruit and vegetables, as well as the alkaline effect of calcium salt. Similar effects were demonstrated in multiple studies [39,64].

Although large observational studies have shown a reduced blood pressure in patients on VLPD/LPD + KAs [39], where a reduced salt intake from LPD and vasodilating effect of BCAA-KAs [65] have been proposed, this has not occurred in our study.

A reduction in triglyceride levels among the participants in the intervention group was also observed. Previous research on diabetic CKD patients has consistently demonstrated that the combination of low-protein diets (LPD) with ketogenic amino acid (KAs) supplementation leads to better fasting serum glucose levels when compared to LPD alone [17]. However, the mechanisms underlying this beneficial effect remain unclear, and it is uncertain whether it is due to reduced carbohydrate intake or improved control of uremia [66]. Recent studies have shown that supplementation of KAs, even with similar protein intake, can lower glycated hemoglobin levels and visceral body fat, and improve lipid metabolism [67]. Based on our findings, we can extrapolate that the reduction in triglyceride levels in the intervention group may be attributed to improved insulin resistance. Further investigation is needed to clarify the mechanisms of action of KA supplementation in CKD patients.

After adjusting for baseline covariates with multivariate regression analysis, which included age, SBP, DBP, Na, and albumin, reduced IS and PCS with increased FMD persisted, which supports our hypothesis that the addition of KAs to LPD improves endothelial function and reduces IS and PCS in CKD stage 3b–4 patients.

However, this study has several limitations. First, it was a single-center study with a limited sample size, which undermines its validity and power. Moreover, this study’s observational and retrospective nature has inherent limitations in attributing improved outcomes to KA supplementation. At the same time, we cannot decipher whether the improvement observed in FMD is due to improved kidney function, solely due to a reduction in uremic toxins, or both. Second, it should be noted that although all patients were being treated with either ACEI or ARBs, the use of alternative antihypertensive agents was not documented. Previous research has indicated that endothelial function is compromised as blood pressure rises, and the extent of dysfunction is directly correlated to the degree of blood pressure elevation [68,69]. Additionally, various studies have demonstrated that endothelial dysfunction is linked to an increase in reactive oxygen species (ROS) in both animal models of hypertension and hypertensive patients [70,71]. In hypertensive rats, for example, nicotinamide adenine dinucleotide phosphate (NADPH) oxidase, a major source of ROS production in vessel walls, is activated [72,73]. These findings suggest that heightened ROS production may contribute to endothelial dysfunction in hypertensive patients. As a result, blood pressure was considered a covariate in the multivariate regression analysis to control for potential confounding in our study. ACEI and ARBS have been demonstrated to reduce the proinflammatory and pro-fibrotic effects of angiotensin II, improving endothelial function and reducing oxidative stress [74]. Other antihypertensive agents recommended in clinical practice have also shown vascular protective benefits. For instance, mineralocorticoid receptor antagonists have been found to lower arterial stiffness and enhance endothelial function, as assessed by flow-mediated dilation [75]. Similarly, calcium channel blockers have demonstrated improvement of endothelial function and reduction of central aortic pressures [76]. Third, we did not address other well-known uremic toxins affecting endothelial function, such as asymmetric dimethylarginine (ADMA) and advanced glycation end products (AGEs). In vitro experimentation has demonstrated that ADMA treatment of endothelial cells leads to apoptosis [77]. At the same time, ADMA is also an endogenous competitive inhibitor of NO synthase [78], and thus may result in endothelial dysfunction. The AGE-RAGE pathway can cause endothelial dysfunction when AGE products bind to the transmembrane receptor (RAGE). This binding triggers a pro-inflammatory and pro-oxidative effect in CKD, ultimately promoting endothelial dysfunction [44]. Fourth, patient adherence to LPD was not ascertained as there was a lack of active monitoring of nitrogen balance. Research has compared CKD patients following LPD versus VLPD + KAs, and the findings suggest that the latter group exhibits higher efficiency of protein turnover [79]. Fifth, although no significant differences in triglyceride (TG) levels were found between the study group and the control group at baseline, our study failed to document other lipid profiles, such as cholesterol and free fatty acid levels. Hyperlipidemia has been identified as an independent cause of endothelial dysfunction. The excessive amounts of lipids, including TGs, free fatty acids, and low-density lipoprotein cholesterol (LDL-C), can cause damage to vascular tissues and their functions, a phenomenon known as lipotoxicity [80]. Lipotoxicity-induced endothelial dysfunction occurs through diverse mechanisms, including increased oxidative stress and proinflammatory responses [81]. Research on middle-aged healthy subjects indicates that impaired FMD is strongly related to the extent of postprandial hypertriglyceridemia [82]. Non-high-density lipoprotein cholesterol (non-HDL-C) also contributes to the impairment of NO bioavailability by increasing the production of reactive oxygen species and inhibiting eNOS activity, leading to endothelial dysfunction and atherosclerosis [83]. Treatment with insulin in type 2 diabetes mellitus and lipid-lowering measures, such as dietary changes or pharmacologic interventions such as statins, restores LDL catabolism and improves endothelial function [84]. Lastly, the absence of hard endpoints, such as major adverse cardiovascular events or time to dialysis, do not sufficiently validate our argument for the effect of restored endothelial dysfunction.

## 5. Conclusions

In conclusion, our study indicates that by supplementing KAs to CKD stage 3b and 4 patients who were on LPD, a decrease in uremic toxin level and improvement in FMD and metabolic profiles was observed in 6 months. These may provide additional benefits, such as delayed CKD progression and decreased risk of cardiovascular disease, without compromising nutritional status. However, further prospective randomized controlled trials are warranted to consolidate this argument, possibly extending the study population to earlier CKD and End Stage Renal Diseases. Hopefully, this could serve as a pilot study for the ongoing investigation of the relationships between KAs, uremic toxins, and endothelial function.

## Figures and Tables

**Figure 1 biomedicines-11-01312-f001:**
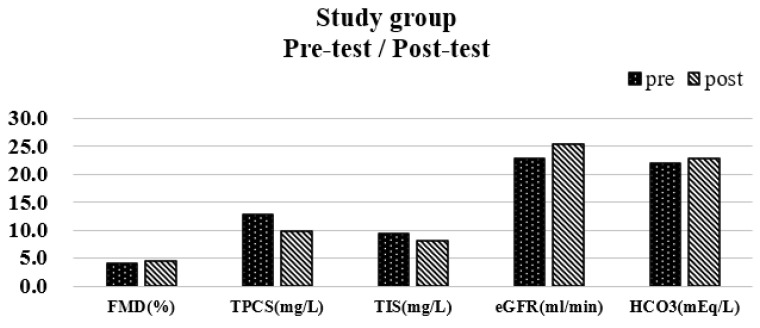
Variables with a significance before and after 6 months in the study group.

**Figure 2 biomedicines-11-01312-f002:**
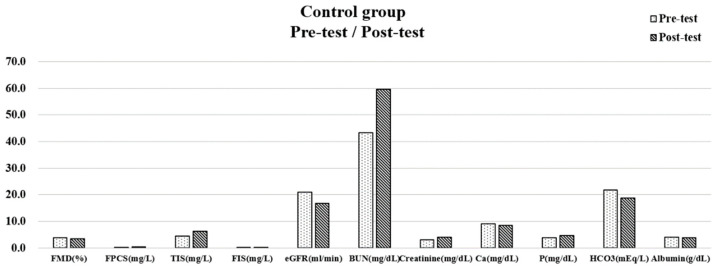
Variables with a significance before and after 6 months in the control group.

**Figure 3 biomedicines-11-01312-f003:**
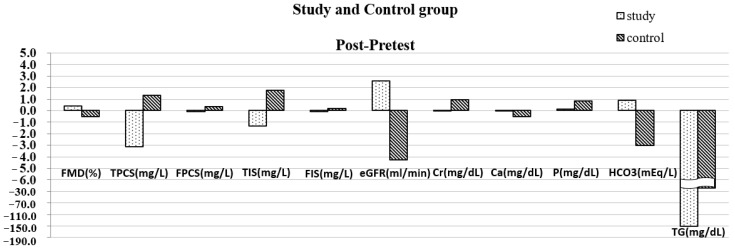
Comparing variables between study and control group after 6 months.

**Table 1 biomedicines-11-01312-t001:** Baseline characteristics of CKD patients who were on LPD for the past 6 months.

Parameters	Study Group, Mean ± SD (n = 11)	Control Group, Mean ± SD (n = 11)	*p*
gender (male)	5 (45.5%)	8 (72.7%)	0.387
age (yrs)	50.36 ± 8.30	57.82 ± 7.28	0.037
FMD (%)	4.16 ± 0.79	3.87 ± 0.40	0.291
TPCS (mg/L)	12.84 ± 14.54	7.65 ± 4.63	0.281
FPCS (mg/L)	0.42 ± 0.78	0.09 ± 0.10	0.191
TIS (mg/L)	9.51 ± 9.82	4.53 ± 2.21	0.129
FIS (mg/L)	0.23 ± 0.31	0.10 ± 0.07	0.175
SBP (mmHg)	127.91 ± 8.26	143.64 ± 4.48	*p* < 0.001
DBP (mmHg)	73.82 ± 5.95	86.55 ± 4.52	*p* < 0.001
eGFR (mL/min)	22.83 ± 8.73	20.97 ± 3.71	0.524
HB (g/dL)	13.14 ± 3.45	10.62 ± 1.37	0.046
Hct (%)	35.55 ± 7.26	31.49 ± 3.46	0.132
BUN (mg/dL)	34.50 ± 8.89	43.40 ± 12.38	0.081
Creatinine (mg/dL)	2.84 ± 1.02	3.04 ± 0.42	0.554
Na (mEq/L)	141.38 ± 1.41	139.09 ± 2.21	0.020
K (mEq/L)	4.40 ± 0.69	4.72 ± 0.36	0.261
Ca (mg/dL)	9.27 ± 0.44	9.01 ± 0.48	0.227
P (mg/dL)	4.27 ± 0.49	3.79 ± 0.61	0.064
HCO3 (mEq/L)	21.96 ± 2.06	21.73 ± 2.05	0.790
Albumin (g/dL)	4.37 ± 0.48	4.02 ± 0.24	0.044
Cholesterol (mg/dL)	162.20 ± 22.44	177.50 ± 54.37	0.562
Triglyceride (mg/dL)	242.60 ± 70.40	178.64 ± 63.48	0.092

FMD, flow-mediated dilatation; TPCS, total p-cresyl sulfate; FPCS, free p-cresyl sulfate; TIS, total indoxyl-sulfate; FIS, free indoxyl sulfate; SBP, systolic blood pressure; DBP, diastolic blood pressure.

**Table 2 biomedicines-11-01312-t002:** Results for primary and secondary outcome after 6 months for both control and study group.

	Study Group			Control Group			Post-Pre Test		
Parameters	Pre-Test, Mean ± SD	Post-Test, Mean ± SD	*p*	Pre-Test, Mean ± SD	Post-Test, Mean ± SD	*p*	Study Group, Mean ± SD	Control Group, Mean ± SD	*p*
FMD (%)	4.16 ± 0.79	4.54 ± 0.64	0.045	3.87 ± 0.40	3.38 ± 0.43	0.001	0.38 ± 0.56	−0.50 ± 0.38	*p* < 0.001
TPCS (mg/L)	12.84 ± 14.54	9.73 ± 11.03	0.020	7.65 ± 4.63	8.95 ± 4.96	0.057	−3.11 ± 3.73	1.31 ± 2.01	0.002
FPCS (mg/L)	0.42 ± 0.78	0.33 ± 0.58	0.191	0.09 ± 0.10	0.45 ± 0.34	0.006	−0.09 ± 0.21	0.36 ± 0.35	0.002
TIS (mg/L)	9.51 ± 9.82	8.15 ± 9.69	0.027	4.53 ± 2.21	6.32 ± 1.91	0.003	−1.36 ± 1.74	1.79 ± 1.50	*p* < 0.001
FIS (mg/L)	0.23 ± 0.31	0.16 ± 0.22	0.104	0.10 ± 0.07	0.29 ± 0.13	0.001	−0.07 ± 0.13	0.20 ± 0.14	*p* < 0.001
SBP (mmHg)	127.91 ± 8.26	128.00 ± 11.66	0.968	143.64 ± 4.48	143.82 ± 7.35	0.921	0.09 ± 7.41	0.18 ± 5.93	0.975
DBP (mmHg)	73.82 ± 5.95	76.36 ± 5.78	0.295	86.55 ± 4.52	89.00 ± 3.07	0.121	2.55 ± 7.65	2.45 ± 4.80	0.974
eGFR (mL/min)	22.83 ± 8.73	25.42 ± 11.28	0.037	20.97 ± 3.71	16.69 ± 3.96	0.001	2.59 ± 3.57	−4.28 ± 3.03	*p* < 0.001
HB (g/dL)	13.14 ± 3.45	12.52 ± 1.93	0.385	10.62 ± 1.37	9.83 ± 1.38	0.056	−1.24 ± 4.06	−0.67 ± 0.96	0.669
Hct (%)	35.55 ± 7.26	36.31 ± 6.95	0.708	31.49 ± 3.46	29.17 ± 3.58	0.060	0.37 ± 3.03	−1.82 ± 2.67	0.104
BUN (mg/dL)	34.50 ± 8.89	35.18 ± 16.50	0.477	43.40 ± 12.38	59.60 ± 12.75	0.013	2.90 ± 12.36	13.78 ± 13.04	0.079
Creatinine (mg/dL)	2.84 ± 1.02	2.80 ± 1.42	0.818	3.04 ± 0.42	4.01 ± 1.11	0.013	−0.04 ± 0.51	0.94 ± 0.96	0.008
Na (mEq/L)	141.38 ± 1.41	140.70 ± 1.16	0.222	139.09 ± 2.21	137.20 ± 3.39	0.252	−0.75 ± 1.58	−1.90 ± 4.91	0.500
K (mEq/L)	4.40 ± 0.69	4.67 ± 0.64	0.197	4.72 ± 0.36	4.98 ± 0.61	0.234	0.15 ± 0.30	0.26 ± 0.64	0.640
Ca (mg/dL)	9.27 ± 0.44	9.25 ± 0.43	0.909	9.01 ± 0.48	8.48 ± 0.38	0.001	−0.01 ± 0.27	−0.50 ± 0.30	0.002
P (mg/dL)	4.27 ± 0.49	4.33 ± 0.77	0.609	3.79 ± 0.61	4.73 ± 0.54	0.002	0.13 ± 0.78	0.84 ± 0.63	0.037
HCO3 (mEq/L)	21.96 ± 2.06	22.85 ± 1.59	0.003	21.73 ± 2.05	18.73 ± 3.04	*p* < 0.001	0.88 ± 0.75	−3.00 ± 1.61	*p* < 0.001
Albumin (g/dL)	4.37 ± 0.48	4.42 ± 0.19	0.669	4.02 ± 0.24	3.79 ± 0.27	0.002	0.06 ± 0.43	−0.23 ± 0.18	0.056
Choleterol (mg/dL)	162.20 ± 22.44	213.67 ± 58.36	0.856	177.50 ± 54.37	199.60 ± 33.85	0.280	1.50 ± 9.19	18.78 ± 48.61	0.642
Trglyceide (mg/dL)	242.60 ± 70.40	98.83 ± 37.99	0.071	178.64 ± 63.48	151.09 ± 34.99	0.260	−188.00 ± 29.70	−27.55 ± 76.51	0.016

**Table 3 biomedicines-11-01312-t003:** Hierarchical multivariate regression analysis (stepwise method) examining group effect.

Parameters	Adjusted For Baseline and Age			Adjusted for Baseline, Age, SBP, Na, and Albumin			Adjusted for Baseline, Age, SBP, DBP, Na, and Albumin		
	B	95% C.I.	*p*	B	95% C.I.	*p*	B	95% C.I.	*p*
FMD (%)	1.00	0.62~1.38	<0.001	0.96	0.50~1.41	<0.001	0.96	0.50~1.41	<0.001
TPCS (mg/L)	−3.25	−4.91~−1.59	<0.001	−3.01	−4.79~−1.22	<0.001			
FPCS (mg/L)	−0.36	−0.60~−0.13	<0.001	−0.37	−0.65~−0.09	0.012	−0.37	−0.65~−0.09	0.012
TIS (mg/L)	−2.92	−4.47~−1.37	<0.001	−3.23	−4.61~−1.84	<0.001	−3.23	−4.61~−1.84	<0.001
FIS (mg/L)	−0.22	−0.33~−0.11	<0.001						

reference group: control group.

## Data Availability

The data presented in this study are available upon reasonable request from the corresponding author.

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
