# Peer review of "Effect of Low Protein Diet Supplemented with Ketoanalogs on Endothelial Function and Protein-Bound Uremic Toxins in Patients with Chronic Kidney Disease"

_biomedicines, 2023, doi:10.3390/biomedicines11051312_

Round 1

Reviewer 1 Report

The present study of Chang and coworkers entitled “Effect of low protein diet supplemented with ketoanalogs on 2 endothelial function and protein-bound uremic toxins in pa-3 tients with chronic kidney disease” reports whether patients suffering from chronic kidney disease benefit from the treatment with ketoanalogs combined to a low protein diet.

The manuscript is well written and organized. The results and methods are clearly described. The discussion section is supporting the reader with comprehensive information and builds up a context about the findings in the field.

The results, which points to an improvement of CKD treatment are interesting for a broad readership of your esteemed journal.

Therefore, I would recommend publication of the present manuscript in your journal.

In the introduction section I noticed some mistakes regarding punctuation:

#p1 lines 38/40 gap “hypertension”

#p1 line 42 space “studies”

#p2 line 58 full stop missing

Best regards

The manuscript is readable. No improvement is needed.

Author Response

Dear Reviewer
Thank you for taking the time to review our manuscript entitled "Effect of low protein diet supplemented with ketoanalogs on endothelial function and protein-bound uremic toxins in patients with chronic kidney disease". We appreciate your feedback ,  mistakes regarding punctuation will be corrected in the revision of manuscript

Thank you again for your time

George Chang

Reviewer 2 Report

Chang G and coworkers are reporting a study in which they explored the effects of a low protein diet (LPD) supplemented or not with ketoaminoacids (KA) on kidney function, protein bound uremic toxins (Indoxyl Sulfate and ParaCresyl Sulfate as total/free fraction) levels and endothelial function (flow mediated dilation (FMD) in advanced chronic kidney disease patients. For this purpose, they conducted a retrospective cohort study, including 22 CKD3-4 patients divided in two groups of 11 patients that received, either a conservative treatment with LPD alone (11) or LPD plus KAs (11) supplementation and were followed up for 6 months. Measurements were performed at baseline and 6 months later for all selected parameters. They showed that after 6 months, LPD with KAs supplementation compared to LPD alone was beneficial, in decreasing uremic toxin levels (IS and PCS) and preserving kidney function, improving FMD and metabolic profiles. These findings tend to reduce risk of cardiovascular disease, without compromising nutritional status.

This is a very interesting and highly clinically relevant study showing additional benefits of ketoamino acid supplementation to low protein diet in advanced CKD patients. Aside the better preservation of kidney function, KAs supplementation tends to reduce protein bound uremic toxins levels and to improve endothelial function better than LPD alone suggesting potential additional benefits of KAs. 

Main limitations of the study are the limited number of patients and the non-randomized aspect of the selection of patients.

Few additional concerns that should be addressed :

1.     At baseline Hb levels are significantly different between the two groups. In other words, anemia management tends to be different between the two groups. Therefore, more information is needed on ESA and/or IV iron supplementation and on kidney disease. It is well known that anemia and ESA are strongly linked to endothelial dysfunction.

2.     Medications are not listed in particular antihypertensive ones. It is also known that RAAS blockers but also blood pressure levels may have an impact on endothelial dysfunction. They should be given

3.     Effects of LPD with or without KAs supplementation on urea generation rate based on 24hr urea mass collected within urine would be of tremendous importance to differentiate between the two LPD approaches on protein catabolism.

4.     Information on diet survey would be also needed to know what are the diet caloric and protein intakes really ingested.

5.     Finally, a detailed information on primary kidney disease and comorbid conditions (diabetes) would also be interesting to compare the two groups.
